# Interleukin 6 (IL-6) Regulates GABAA Receptors in the Dorsomedial Hypothalamus Nucleus (DMH) through Activation of the JAK/STAT Pathway to Affect Heart Rate Variability in Stressed Rats

**DOI:** 10.3390/ijms241612985

**Published:** 2023-08-19

**Authors:** Lihua Zhang, Weibo Shi, Jingmin Liu, Ke Chen, Guowei Zhang, Shengnan Zhang, Bin Cong, Yingmin Li

**Affiliations:** Hebei Key Laboratory of Forensic Medicine, Collaborative Innovation Center of Forensic Medical Molecular Identification, Department of Forensic Medicine, Hebei Medical University, Shijiazhuang 050017, China; zhanglihua0622@126.com (L.Z.); 18401452@hebmu.edu.cn (W.S.); m18731868637@163.com (J.L.); ckatjia955@163.com (K.C.); zguowei1020@126.com (G.Z.); zhangsn1211@163.com (S.Z.)

**Keywords:** stress, dorsomedial hypothalamus nucleus, JAK/STAT, GABAA receptor, heart rate

## Abstract

The dorsomedial hypothalamus nucleus (DMH) is an important component of the autonomic nervous system and plays a critical role in regulating the sympathetic outputs of the heart. Stress alters the neuronal activity of the DMH, affecting sympathetic outputs and triggering heart rate variability. However, the specific molecular mechanisms behind stress leading to abnormal DMH neuronal activity have still not been fully elucidated. Therefore, in the present study, we successfully constructed a stressed rat model and used it to investigate the potential molecular mechanisms by which IL-6 regulates GABAA receptors in the DMH through activation of the JAK/STAT pathway and thus affects heart rate variability in rats. By detecting the c-Fos expression of neurons in the DMH and electrocardiogram (ECG) changes in rats, we clarified the relationship between abnormal DMH neuronal activity and heart rate variability in stressed rats. Then, using ELISA, immunohistochemical staining, Western blotting, RT-qPCR, and RNAscope, we further explored the correlation between the IL-6/JAK/STAT signaling pathway and GABAA receptors. The data showed that an increase in IL-6 induced by stress inhibited GABAA receptors in DMH neurons by activating the JAK/STAT signaling pathway, while specific inhibition of the JAK/STAT signaling pathway using AG490 obviously reduced DMH neuronal activity and improved heart rate variability in rats. These findings suggest that IL-6 regulates the expression of GABAA receptors via the activation of the JAK/STAT pathway in the DMH, which may be an important cause of heart rate variability in stressed rats.

## 1. Introduction

Stress is closely related to cardiac function [1]. Statistical studies have shown that the incidence of acute coronary syndrome in the Munich area increased significantly by a factor of 2.66 on the day of the 2006 World Cup match for Germany, particularly the 2 h after the matches started [2]. In addition, Kivimäki et al. also found that employees exposed to chronic stress in the workplace have, on average, a 50% higher risk of coronary heart disease [3]. Stress elicits increases in sympathetic and hypothalamic–pituitary–adrenocortical axis activity, resulting in an increased heart rate and blood pressure, thus providing adequate hemodynamic and metabolic support to enhance the body’s ability to respond to various stimuli [4]. However, excessive stress leads to adverse adaptive reactions in the body [5,6], especially when pathological changes occur in some key brain regions that regulate autonomic and endocrine functions. These negative consequences on the body could form a continuous “vicious cycle.” The dorsomedial hypothalamus nucleus (DMH) is a key brain region that integrates the autonomic, endocrine, and behavioral responses to stress, and plays a critical role in maintaining the homeostasis of the internal environment of the organism [7]. The DMH is located caudal and ventral to the paraventricular nucleus of the hypothalamus, and lateral to the fornix and lateral hypothalamic area. It is directly responsible for regulating the sympathetic outputs of stress on the cardiovascular system [8,9]. Previous studies have shown that stress stimulation led to abnormal DMH neuron activity and tachycardia, while specific inhibition of DMH neurons significantly reduced this heart rate variability [10,11], indicating that the abnormal activity of DMH neurons is an important cause of stress-induced tachycardia in rats. However, the molecular mechanisms underlying the abnormal activity of DMH neurons caused by stress are currently unclear.

The γ-aminobutyric acid type A receptor (GABAAR) is a heteropentameric receptor belonging to the ligand-gated ion channel superfamily that mediates most of the inhibitory synaptic transmission of GABA in the central nervous system [12]. Activation of GABAAR mediates chloride flow into the cell, leading to neuronal hyperpolarization, which, in turn, inhibits neuronal activity. Studies have shown that GABAAR is very sensitive to subtle changes in the environment [13,14], and its dynamic changes strongly affect the maintenance of neuronal homeostasis, which is closely related to the development of various neurological diseases, such as epilepsy, depression, and Alzheimer’s disease [15,16]. With the development of technology, GABAAR now has 19 identified subunits (α1-6, β1-3, γ1-3, δ, ε, π, θ and ρ1-3, etc.), among which, GABAA receptors containing the α1 subunit are more abundantly expressed in the mammalian nervous system, accounting for approximately 60% of all GABAA receptors [17,18]. The results of numerous studies on ischemic stroke and traumatic brain injury have suggested that reduced expression of the GABAAR α1 subunit at the site of injury was an important cause of enhanced neuronal excitability leading to the development of epilepsy [19,20,21]. Therefore, these important findings suggest that abnormal expression of the GABAAR α1 subunit may be involved in the tachycardia triggered by increased neuronal activity of the DMH in stressed rats.

As an important pro-inflammatory cytokine, interleukin 6 (IL-6) has an important role in the development of stress-related diseases [22,23]. IL-6 is a classical extracellular stimulator of the Janus kinase/signal transducer and activator of transcription (JAK/STAT) signaling pathway. It activates the JAK/STAT pathway by binding to the transmembrane protein gp130 to initiate intracellular signaling [24,25]. The JAK/STAT pathway is closely associated with neuronal proliferation, survival, development, and differentiation, and also affects long-range enhancement and long-range inhibition of neuronal signals [26]. Recent studies have found that activation of the JAK/STAT pathway in models of epilepsy and traumatic brain injury could regulate the expression of GABAAR by increasing the levels of phosphorylated STAT3 (pSTAT3). This upregulates the downstream molecule known as the inducible cAMP early repressor (ICER), which is a member of the cyclic AMP response element-binding protein (CREB) family [27,28]. However, it is not clear whether IL-6 is abnormal under stress and regulates the expression of the GABAAR α1 subunit by activating the JAK/STAT pathway.

Based on the above information, we proposed the hypothesis that stress-induced IL-6 activates the JAK/STAT pathway to downregulate the expression of the GABAAR α1 subunit in the DMH, which enhances neuronal activity and triggers tachycardia. In order to verify our hypothesis, we examined the dynamic changes in heart rate, neuronal activity of the DMH, and factors related to the JAK/STAT pathway using a stressed rat model. This allowed us to elucidate the potential molecular mechanism of stress leading to the decrease in the GABAAR α1 subunit and its effect on the neuronal activity of the DMH and triggering a change in heart rate.

## 2. Results

### 2.1. Stress-Induced Enhancement of DMH Neuronal Activity Increased Heart Rate

The c-Fos gene is one of the most widely studied immediate early genes in the brain and is expressed when neurons are excited, which often serves as a marker of neuronal activation. Previous studies have shown that stress could activate DMH neurons and increase the heart rate in rats [8,9]. Therefore, in this study, we first detected the changes in c-Fos protein expression in the DMH and heart rate of rats by immunohistochemical staining and electrocardiography, respectively. Quantitative analysis of the immunohistochemical staining revealed that stress treatment led to a significant effect on the expression of c-Fos (Figure 1A,B). Post hoc testing indicated that the percentage of c-Fos-positive cells obviously increased after 3 days (*p* < 0.01), 7 days (*p* < 0.01), and 21 days (*p* < 0.01) of stress exposure compared with the control group. This suggests that stress significantly activates DMH neurons. Given that the activation of DMH neurons is closely associated with an increase in heart rate, we subsequently monitored the changes in HR (beats/min, bpm) of rats after the stress treatment. ANOVA and post hoc comparisons demonstrated that the HRs of the rats were significantly increased after 3 days (*p* < 0.01), 7 days (*p* < 0.01), and 21 days (*p* < 0.01) of stress exposure (Figure 1C,D).

### 2.2. Stress Decreased the Expression of the GABAAR α1 Subunit

In the central nervous system, the GABAAR α1 subunit mediates inhibitory synaptic transmission to suppress neuronal excitability. In order to investigate whether stress exposure alters the GABAAR α1 subunit and the activity of DMH neurons, we examined the mRNA and protein levels of the GABAAR α1 subunit. ANOVA and post hoc testing indicated that the mRNA levels of the GABAAR α1 subunit decreased significantly after 3 days (*p* < 0.01), 7 days (*p* < 0.01), and 21 days (*p* < 0.01) of stress exposure compared with the control group (Figure 2A). The change in the GABAAR α1 subunit protein expression was paralleled by a similar effect in transcript levels. Quantitative analysis of the immunohistochemistry and Western blot results both showed that the protein expression of the GABAAR α1 subunit was significantly decreased after 3 days (*p* < 0.05), 7 days (*p* < 0.01), and 21 days (*p* < 0.01) of stress exposure compared with the control group (Figure 2B–E). These data suggest that stress-induced reduction in GABAAR α1 subunit expression is associated with enhanced DMH neuronal activity.

### 2.3. Stress-Induced Increase in IL-6 Activated the JAK/STAT Pathway, Promoting the Expression of the Downstream Molecule ICER

Studies have shown that stress-induced IL-6 can reduce the GABAergic postsynaptic current amplitude in the temporal cortex by inhibiting GABAA receptor expression, thus enhancing neuronal excitability [29,30]. To investigate whether IL-6 was elevated under exposure to stress, we examined IL-6 levels in the DMH by ELISA and immunohistochemistry. ANOVA and post hoc comparisons showed that IL-6 levels in the DMH were significantly enhanced after 3 days (*p* < 0.01), 7 days (*p* < 0.05), and 21 days (*p* < 0.05) of stress exposure (Figure 3A). Quantitative analysis of the immunohistochemistry also showed that stress had a significant effect on the expression of IL-6 protein (Figure 3B,C). Post hoc testing indicated that the expression of IL-6 was significantly increased after 3 days (*p* < 0.05), 7 days (*p* < 0.01), and 21 days (*p* < 0.01) of stress exposure compared with the control group. To further investigate whether the stress-induced increase in IL-6 expression activated the JAK/STAT pathway to regulate the GABAA receptor α1 subunit, we detected the protein levels of pSTAT3, a key protein of the JAK/STAT pathway. Both immunohistochemistry and Western blot analysis displayed a significant difference in pSTAT3 protein levels after stress exposure (Figure 3D–G). Post hoc testing showed that the level of pSTAT3 gradually increased after 3 days (*p* < 0.05), 7 days (*p* < 0.05), and 21 days (*p* < 0.01) of stress exposure compared with the control group.

It has been reported in previous studies that the activation of the JAK/STAT pathway could significantly increase the transcriptional and translational levels of ICER [27], which regulates the expression of the GABAAR a1 subunit. We, therefore, investigated the effect of stress on ICER expression. As is shown in Figure 3H, there was an obvious increase in ICER mRNA after stress exposure compared with the control group (*p* < 0.05). The change in ICER protein expression was paralleled by a similar effect in terms of the mRNA level, as measured by immunohistochemistry and Western blot analysis (Figure 3I–L). Post hoc testing indicated that the protein expression of ICER was obviously increased after 3 days (*p* < 0.01), 7 days (*p* < 0.01), and 21 days (*p* < 0.05) of stress exposure compared with the control group. The cAMP response element-binding protein (CREB) is a transcription factor that regulates the transcription of target genes through the phosphorylation of Ser-133 in response to various stimuli. Studies have shown that pCREB is an important regulator motor of the GABAAR α1 subunit to inhibit its transcription. Therefore, we subsequently examined the protein levels of pCREB. As was shown by immunohistochemical staining, the stress treatment resulted in a significant difference in the pCREB protein levels (Figure 3M–N). Post hoc testing indicated that the protein expression of pCREB was significantly increased after 3 days (*p* < 0.01), 7 days (*p* < 0.05), and 21 days (*p* < 0.05) of stress exposure compared with the control group. These results illustrated that the stress-induced increase in IL-6 levels activated the JAK/STAT pathway and promoted the expression of the downstream molecule ICER, which is involved in the suppression of GABAAR α1 subunit expression in the DMH of stressed rats.

### 2.4. AG490 Treatment Inhibited the Levels of pSTAT3 and Its Downstream Molecule ICER in Stressed Rats

To further investigate the molecular mechanism by which the JAK/STAT pathway promotes ICER expression to regulate the GABAAR α1 subunit, we inhibited the JAK/STAT pathway of rats stressed for 7 days by lateral ventricular injection of the JAK2-specific inhibitor AG490, which has been shown to significantly inhibit STAT3 phosphorylation. We first examined the protein levels of pSTAT3 by immunohistochemistry and Western blot analysis. There were obvious effects of the interaction of stress and AG490 treatment (*p* < 0.05), stress treatment alone (*p* < 0.01), and AG490 treatment alone (*p* < 0.01) on the protein levels of pSTAT3. Post hoc testing showed that there were significant increases in the protein expression of pSTAT3 (*p* < 0.05) after stress treatment (Figure 4A–D). In contrast, AG490 treatment significantly decreased the protein expression of pSTAT3 in stressed rats (*p* < 0.05). Subsequently, we examined the ICER mRNA and protein levels. The RNAscope results showed that the interaction of stress and AG490 treatment (*p* < 0.01) and stress treatment alone (*p* < 0.01) had significant effects on the level of ICER mRNA. Post hoc testing showed that stress and AG490 treatment significantly decreased ICER mRNA expression compared with stress treatment alone (*p* < 0.05) (Figure 5A,B). The changes in ICER protein expression were in line with the ICER mRNA levels, as measured by immunohistochemical staining (Figure 5C,D) and Western blot analysis (Figure 5E,F). These data suggest that AG490 significantly inhibited the levels of pSTAT3 and its downstream molecule ICER in the DMH of stressed rats.

### 2.5. AG490 Treatment Alleviated the Stress-Induced Decrease in the GABAAR α1 Subunit

Next, we observed the effect of AG490 on the mRNA and protein levels of the GABAAR α1 subunit in stressed rats. As is shown in Figure 6A,B, the RNAscope results indicated significant effects of the interaction of stress and AG490 treatment (*p* < 0.05), stress treatment alone (*p* < 0.05), and AG490 treatment alone (*p* < 0.05) on GABAAR α1 subunit mRNA levels. Post hoc testing showed that there was a decrease in GABAAR α1 subunit mRNA levels after stress treatment (*p* < 0.05). In contrast, AG490 treatment significantly increased the mRNA expression of the GABAAR α1 subunit in the stressed rats (*p* < 0.05). Meanwhile, similar effects were seen at the protein level. Both immunohistochemical staining and Western blot analysis showed that the interaction of stress and AG490 treatment (*p* < 0.01), stress treatment alone (*p* < 0.01), and AG490 treatment alone (*p* < 0.05) had significant effects on the protein expression of the GABAAR α1 subunit. Post hoc testing showed that stress and AG490 treatment led to a significant increase in protein expression of the GABAAR α1 subunit compared with stress treatment alone (*p* < 0.05) (Figure 6C–F). The above results suggest that inhibition of the JAK/STAT pathway alleviated the stress-induced decrease in the expression of the GABAAR α1 subunit.

### 2.6. AG490 Treatment Reduced DMH Neuronal Activity and Improved Tachycardia in Stressed Rats

To further determine whether restoration of GABAA α1 subunit expression could rescue the DMH neuronal activity and HR defects in stressed rats, we used immunohistochemical staining to analyze the percentage of c-Fos-positive cells. As is shown in Figure 7A,B, the interaction of stress and AG490 treatment (*p* < 0.05), stress treatment alone (*p* < 0.01), and AG490 treatment alone (*p* < 0.01) had significant effects on the expression of c-Fos protein. Post hoc testing indicated that stress and AG490 treatment resulted in a significant decrease in the percentage of c-Fos-positive cells (*p* < 0.01) compared with stress treatment alone. Meanwhile, we also investigated the changes in the HR of rats after the administration of stress and AG490 (Figure 7C,D). The interaction of stress and AG490 treatment (*p* < 0.05), stress treatment alone (*p* < 0.01), and the AG490 treatment alone (*p* < 0.05) had significant effects on the change in HR. Post hoc testing indicated that HR increased obviously after stress treatment alone (*p* < 0.01). In contrast, stress and AG490 treatment significantly decreased the HR of rats (*p* < 0.05). These above results indicate that AG490 can reduce DMH neuronal activity and improve tachycardia by restoring GABAAR α1 subunit expression in stressed rats.

## 3. Discussion

Stress refers to the series of neuroendocrine, physiological, and pathological responses that occur when the body is stimulated by various stressors. Stress-induced dysfunction of the sympathetic–adrenomedullary axis is an important cause of tachycardia and the subsequent series of adverse consequences [10]. The DMH is an important relay station of the sympathetic nervous system, which receives the extensive integration of projections from the cortex, amygdala, and other limbic systems, and then projects directly or via the periaqueductal gray (PAG) to sympathetic preganglionic neurons in the raphe pallidus (RPa) to regulate the heart rate [9,31]. It has been well documented that stress-induced enhancement of DMH neuronal activity is responsible for tachycardia in rats, and the inhibition of DMH neuronal activity during stress could effectively reduce the occurrence of these adverse consequences [9,32]. Therefore, to investigate the detailed mechanism by which stress leads to enhanced DMH neuronal activity and thus affects cardiac rhythm, we established stressed rat models using different durations of restraint-related stress, plus ice water swimming, and used this model to examine the activity of DMH neurons and changes in heart rate. The results showed that stress significantly activated DMH neurons leading to tachycardia in rats, which was consistent with previous reports. However, it was noteworthy that the significant increase in heart rate with prolonged stress exposure was not completely consistent with the activation of DMH neurons. This suggests that the effect of chronic stress on cardiac function may be influenced by other endocrine effects, in addition to the direct sympathetic regulation by DMH neurons.

The activation of DMH neurons is regulated by excitatory and inhibitory neurotransmitters released from upstream nerve fibers acting on their receptors. Previous studies have suggested that the abnormal activity in DMH brain regions during stress may be mainly due to a diminished inhibitory effect on these neurons [33]. GABAAR containing the α1 subunit is widely expressed as an inhibitory receptor in the mammalian nervous system [34]; Lund IV et al. indicated that the reduced expression of the GABAAR α1 subunit was an important cause of epilepsy triggered by the hyperexcitability of rat hippocampal neurons [35,36]. Therefore, we hypothesized that stress could lead to the abnormal expression of the GABAARα1 subunit in DMH neurons and thus directly affect neuronal activity. We then examined the expression of the GABAAR α1 subunit at the mRNA and protein levels, and the data indicated that the expression of the GABAAR α1 subunit decreased significantly with the prolongation of stress. This suggests that the reduced expression of the GABAAR α1 subunit caused by stress was an important reason for the abnormal increase in neuronal activity in the DMH.

Stress increases the levels of various pro-inflammatory cytokines in the body. IL-6, an important pro-inflammatory factor, plays a critical role in cell proliferation, differentiation, survival, and apoptosis, in addition to participating in the regulation of neuronal activity [37,38]. It has been demonstrated that IL-6 reduces the amplitude of GABAergic postsynaptic currents in the temporal cortex through the inhibition of GABAA receptors, thus enhancing neuronal activity [29,30]. Therefore, we examined the level of IL-6 in stressed rats, and the results showed that stress significantly increased the level of IL-6 in the DMH. These data suggest that stress enhances neuronal activity by inhibiting GABAA receptors through increased levels of IL-6. Previous studies have shown that increased expression of IL-6 could significantly activate the JAK/STAT pathway to initiate intracellular signaling. The JAK/STAT pathway mainly consists of tyrosine-kinase receptors, JAKs (JAK1, JAK2, JAK3, and TYK2), and STATs (STAT1, STAT2, STAT3, STAT4, STAT5A, STAT5B, and STAT6). Of these, JAK2 and STAT3 are highly expressed in the central nervous system and play an important role in the regulation of long-range enhancement and long-range inhibition of neuronal signaling [26,39]. The binding of IL-6 to its receptor causes the activation and phosphorylation of JAK2, which is coupled to the receptor, and leads to the phosphorylation of tyrosine residues of the receptor and the formation of STAT docking sites. STAT3, through its SH2 structural domain, is recruited to the docking sites and is then phosphorylated in the presence of pJAK2. Subsequently, pSTAT3 dissociates from the receptor and forms homo- or heterodimers via intermolecular SH2 structural domain–phosphotyrosine interactions. These, in turn, translocate to the nucleus to bind DNA regulatory elements to regulate the transcription and translation of downstream target genes [27,40]. Since the promoter of ICER is one of the target loci regulated by pSTAT3, the activation of the JAK2/STAT3 pathway could significantly increase the transcription and translation levels of ICER [27,36]. It has been demonstrated that the overexpression of ICER significantly downregulated the GABAAR α1 subunit. Our study confirmed that stress-induced IL-6 significantly increased the expression of pSTAT3 and increased the mRNA and protein levels of the downstream factor ICER. CREB is a transcription factor that is phosphorylated at Ser-133 in response to various stimuli and regulates the transcription of target genes which are involved in the regulation of neuronal survival, development, excitability, and synaptic plasticity [41]. It has been shown that pCREB is an important factor in the regulation of the GABAAR α1 subunit expression. It can form a heterodimer with ICER and bind to the promoter of the GABAAR α1 subunit, thus inhibiting the transcription of the GABAAR α1 subunit [27,36]. Our results indicated that the expression of pCREB increased significantly along with the increase in ICER expression as the duration of stress increased. This indicates that the binding of pCREB and ICER to the promoter of GABAAR α1 was enhanced, which in turn, inhibited the transcription of the GABAAR α1 subunit. In addition, it was noteworthy that the expression level of ICER was not consistently increased with prolonged stress exposure. It is known that pCREB can also regulate ICER to enhance its expression [42]. Therefore, we consider that the increase of pCREB might be involved in upregulating ICER expression and thus causing it to peak at RSIS3. However, ICER, as a transcriptional repressor, could repress its own expression by forming ICER homodimers when its expression reaches a certain level [43]. This may explain why ICER expression was slightly decreased at RSIS7 and RSIS21. In summary, we tentatively suggest that the stress-induced increase in IL-6 expression activates the JAK/STAT pathway, increasing the transcription and translation of the downstream molecule ICER. This process inhibits the expression of the GABAAR α1 subunit, leading to increased DMH neuronal activity and triggering an increased heart rate.

To further verify the above findings, we used a lateral ventricular injection of AG490 to inhibit the JAK/STAT pathway in the DMH of rats stressed for 7 days. AG490 is a specific inhibitor of JAK2 and can significantly inhibit the phosphorylation of STAT3. The literature has reported that AG490 cab effectively ameliorated the central and peripheral inflammatory responses and metabolic abnormalities in rats with ischemic stroke, and that it had neuroprotective effects against cerebral ischemia–reperfusion injury and traumatic brain injury [39,44,45]. Our results indicated that AG490 significantly inhibited the level of pSTAT3 and its downstream factor ICER in the DMH neurons of stressed rats. In addition, the expression of the GABAAR α1 subunit was significantly increased, demonstrating that stress was able to inhibit the GABAA receptor α1 subunit by activating the JAK/STAT pathway to promote the expression of the downstream molecule ICER. As changes in the expression of the GABAAR α1 subunit could directly affect neuronal activity, we further examined the expression of c-Fos in the neurons of the DMH and heart rate changes in rats. The results showed that the activity of DMH neurons in the stressed rats was significantly reduced, and tachycardia was significantly improved after AG490 treatment. Taking these results together, our study confirmed that stress-induced IL-6 activates the JAK/STAT pathway to inhibit GABAAR α1 subunit expression, which, in turn, enhances DMH neuronal activity, triggering tachycardia in rats.

## 4. Materials and Methods

### 4.1. Animals

All procedures were conducted in accordance with the National Institutes of Health guidelines and were approved by the Institutional Review Board for Animal Experiments at Hebei Medical University (IACUC-Hebmu-2023011). Healthy adult male Sprague Dawley (SD) rats (200 ± 20 g) were purchased from Beijing Vital River Laboratory Animal Technology Co., Ltd., China (Beijing, China), and were housed in a temperature- and humidity-controlled room with a 12 h light–12 h dark cycle. Food and water were available ad libitum. The rats were randomly allocated into four groups: the control group and groups exposed to restraint stress combined with ice water swimming (RSIS) for 3, 7, and 21 days (n = 12/group). In addition, to further investigate the molecular mechanisms by which the JAK/STAT pathway regulates the GABAAR α1 subunit, four additional groups of rats were added as follows: vehicle-treated control group (CON + Vehicle); AG490-treated control group (CON + AG490); vehicle-treated stress group (Stress + Vehicle); and AG490-treated stress group (Stress + AG490) (n = 12/group). AG490 (Cayman Chemical, Ann Arbor, MI, USA) is a specific inhibitor of JAK2, which significantly downregulates the level of pSTAT3. AG490 (2 μL, 5 mM, i.c.v.) was dissolved in 3% dimethyl sulfoxide (DMSO) [46,47]. The vehicle-treated groups received the same volume of 3% DMSO.

### 4.2. RSIS Procedure

The RSIS protocols were conducted as per our previously described method [48]. Briefly, rats were placed in a restraint device with no access to food and water for 8 h (from 8:00 a.m. to 4:00 p.m.) each day. Then, the restrained rats were placed in ice water to swim for 5 min each day. The process was repeated for 3, 7, or 21 days. The control group rats were left in their cages for the same amount of time without food or water. Rat body weight and behavioral changes indicated that the stressed rat model was established successfully (Appendix A).

### 4.3. Intraventricular Injection

The rats in the vehicle-treated and AG490-treated groups were anesthetized with 2% pentobarbital sodium (0.3 mL/100 g, i.p.) and fixed in a stereotaxic apparatus. A small longitudinal incision was made on the head to expose the skull, and the bregma was labeled to determine the coordinates. Guide cannulas (RWD Life Science, Shenzhen, China) were implanted bilaterally into the lateral ventricles to allow for intracerebroventricular (i.c.v.) injections. The coordinates of the lateral ventricle were 0.92 mm anteroposterior (AP), 1.5 mm mediolateral (ML), and 3.5 mm dorsoventral (DV) to the bregma. The guide cannulas were fixed to the skull with stainless steel screws and dental cement. Then, the skull was covered with a dust cap to prevent occlusion. The rats were immediately removed from the stereotaxic apparatus and placed on a heating pad to maintain their basal body temperature after surgery. Finally, the animals were injected with penicillin (8U/day, i.p.) for 3 days and were allowed to recover for one week.

The i.c.v. microinjection was performed using a polyethylene tube connected to a microinjection needle (Hamilton, Reno, NV, USA) on one end, and the i.c.v. guide cannulas on the other. Microinjections were performed within a 5 min period. Subsequently, the microinjection needle was left in the guide cannula for longer than 1 min until the injected liquid had completely diffused into each of the lateral ventricles. This procedure was performed 30 min before the daily stress treatment and continued for 7 consecutive days.

### 4.4. Electrocardiogram Measurement

The electrocardiograms of the rats were conducted 3 h after the behavioral experiments had concluded. The electrocardiogram was monitored using the PowerLab system (AD Instruments, Bella Vista, Australia). After being anesthetized with isoflurane, the rats were fixed on the experimental platform, and the platinum electrodes were inserted subcutaneously by way of a lead II electrode connection. The rat ECGs were subsequently recorded using a Bio-Amp. An artifact-free 5-min segment of the ECG was analyzed using LabChart 7.3.8 software.

### 4.5. Enzyme-Linked Immunosorbent Assay (ELISA)

The rats were anesthetized with 2% pentobarbital sodium (0.3 mL/100 g, i.p.), and their brains were then removed. The samples were flash-frozen in liquid nitrogen. DMH tissues were then isolated on ice with the aid of a brain matrix and stored at −80 °C. An appropriate amount of collected DMH tissue was subsequently weighed and homogenized in a 9-fold volume of PBS buffer. The samples were then centrifuged at 3000 rpm for 10 min at 4 °C, and the supernatants were collected for assay. The concentration of IL-6 in the tissues was measured according to the instructions of the Rat IL-6 Uncoated ELISA Kit (Invitrogen, Waltham, MA, USA). Absorbance at 450 nm was measured using a microplate reader (BioTek Epoch, Agilent, Santa Clara, CA, USA).

### 4.6. Real-Time Quantitative Polymerase Chain Reaction (RT-qPCR)

According to the manufacturer’s instructions, total mRNA was extracted from DMH tissue using Trizol (Invitrogen, USA), and then cDNA was synthesized using the RNA-PCR Reverse Transcription Kit (RR047A; Takara Bio, Beijing, China). Quantification of GABAARα1 and ICER mRNA of the rat DMH tissue was carried out via the SYBR Green RT-PCR kit (RR820A; Takara Bio, Beijing, China). Relative expression of the target gene was mainly calculated using the 2^−ΔΔCt^ method with the glyceraldehyde 3-phosphate dehydrogenase (GAPDH) gene as the internal control.

The sequences of the primers used are as follows:

GABAARα1: Forward Primer: 5′-GTATTCAGCTCGGGGACAGG-3′;

Reverse Primer: 5′-GCACACTATATCATTTGTGCGA-3′.

ICER: Forward Primer: 5′-AAGAAGCAACTCGAAAGCGG-3′;

Reverse Primer: 5′-CTGCCCCATTAGAGTTCACAGT-3′;

GAPDH: Forward Primer: 5′-GTCTCCTCTGACTTCAACAGCG-3′;

Reverse Primer: 5′-ACCACCCTGTTGCTGTAGCCAA-3′.

### 4.7. Immunohistochemistry (IHC)

The isolated rat brain was fixed in 10% formalin, and the tissue was subsequently dehydrated in a gradient of ethanol solutions and embedded in paraffin. According to the Rat Brain in Stereotaxic Coordinates, the brain wax block was sectioned in continuous sections (5 μm) on the largest coronal surface of the DMH for immunohistochemical staining. The sections were deparaffinized with xylene and alcohol and incubated in a 0.01 mol/L citrate buffer (pH 6.0) for microwave antigen repair. This was followed by incubation in 3% H_2_O_2_ with methanol for 25 min at room temperature. The tissue sections were blocked in goat serum for 40 min and then incubated with the following primary antibodies overnight at 4 °C: anti-cFos (1:500, Abcam, Cambridge, UK); anti-GABAA receptor α1 (1:500, Santa Cruz, CA, USA); anti-pSTAT3-Tyr705 (pSTAT3) (1:100, HUOBIO, Hangzhou, China); anti-CREM(C-2)/ICER (1:200, Santa Cruz, CA, USA); or anti-pCREB1-S133 (pCREB) (1:100, ABclonal, Wuhan, China). Next, the tissues were incubated with biotinylated secondary antibodies for 40 min at 37 °C, followed by incubation with horseradish peroxidase-conjugated biotin for 40 min at 37 °C. Finally, the sections were incubated with 3,3′-diaminobenzidine tetrahydrochloride (DAB) and were counterstained with hematoxylin.

### 4.8. Analysis of IHC Staining

Digital images of the stained slides were acquired using an Aperio CS2 scanner (Leica Biosystems, Vista, CA, USA) at 40× magnification. After saving the digital images, the DMH regions were selected for analysis using the Aperio ImageScope 12.4.6 software, which has been applied to quantitative analysis of immunohistochemistry with considerable accuracy and repeatability [49,50]. The protein expression was assessed with the Nuclear v9 and Color Deconvolution v9 algorithms of the Aperio ImageScope software. These algorithms are based on spectral differentiation between brown (positive) and blue (counter) staining. The results were reported as the percentage of positive cells and the average positive intensity, respectively.

### 4.9. RNAscope Assay

RNAscope detection was performed as previously described [28]. The procedure included the following steps: After deparaffinization and hydration with xylene and alcohol, the sections were incubated with hydrogen peroxide reagents (ACD, cat. no.3223330) at room temperature for 10 min. This was followed by boiling in target retrieval reagents for 15 min at 99 °C. They were then incubated with protease reagents for 30 min at 40 °C. The following target probes were added to the sections and incubated for 2 h at 40 °C within a humidity control chamber: positive control probe Rn-Ppib (cat. no. 313921), negative control probe DapB (cat. no. 31004311), Rn-Crem-03 (cat. no. 530001), and Rn-Gabral (cat. no. 411601). The sections were then incubated with signal amplification reagents: AMP 1, AMP 2, AMP 3, AMP 4, AMP 5, and AMP 6 reagents for 30, 15, 30, 15, 30, and 15 min, respectively. Finally, the signals were detected using an RNAscope^®^ 2.5HD Detection Reagent-RED kit (ACD, cat. no. 322360). The sections were counterstained with hematoxylin for 2 min at room temperature, followed by visualization using a microscope. For the quantification of RNAscope, the Visiopharm data analysis 2018.4 software was used to count the number of positive signals and cells in the DMH separately, and the average number of positive points per cell was calculated.

### 4.10. Western Blot Analysis

The DMH tissues of rats were homogenized mechanically in RIPA buffer mixed with a protease inhibitor cocktail (RW-0001) and phosphatase inhibitor (RP-WA0130). After centrifugation at 12,000 *g* at 4 °C for 5 min, the protein concentration was measured using a BCA quantification kit (RP-WA0201). Additionally, proteins were run on 8–10% polyacrylamide SDS-PAGE gels for electrophoresis (ZomanBio, ZD304C, Beijing, China), and subsequently transferred to polyvinylidene fluoride (PVDF) membranes using a Bio-Rad Transblot (Sigma-Aldrich, REF03010040001, Saint Louis, Missouri, USA). These membranes were blocked for 2 h in 5% skim milk/TBST or 5% bovine serum albumin/TBST at 37 °C and then incubated with antibodies against GAPDH (1:1000, Beyotime, China), β-tubulin (1:1000, Abcam, Cambridge, UK), GABAAR α1 (1:500, Sigma-Aldrich, Saint Louis, Missouri, USA), pSTAT3 (1:1000, HUABIO, Hangzhou, China), STAT3 (1:1000, HUABIO, Hangzhou, China), and CREM(C-2)/ICER (1:200, Santa Cruz) overnight at 4 °C. Then, the PVDF membranes were incubated with the corresponding fluorescent secondary antibody (LI-COR, 926-68071; Rockland, 610-145-002) at 37 °C, away from light, for 1 h. The signal was captured on an Odyssey dual-color infrared fluorescence imaging system (LI-COR, Lincoln, Nebraska, USA), and the protein bands were quantified and analyzed using ImageJ 1.52a software (NIH, Bethesda, MD, USA).

### 4.11. Statistical Analysis

All experiments were performed independently at least three times. All data are shown as the mean ± SEM and were analyzed using SPSS 25.0 (IBM SPSS Statistics, Chicago, IL, USA). GraphPad Prism version 8.0 (GraphPad Software Inc., CA, USA) was used for graph generation. One-way or two-way analysis of variance (ANOVA) followed by LSD or Tukey’s post hoc test were applied to analyze data for comparisons between groups. A value of *p* < 0.05 was considered significant.

## 5. Conclusions

In this study, we employed conventional pathological, molecular biology, and RNAscope techniques to investigate whether the increased expression of the inflammatory factor IL-6 under stress could regulate the GABAAR α1 subunit by activating the JAK/STAT pathway. We also studied whether this affected the neuronal activity of the DMH and triggered heart rate changes using a rat model of induced stress. The findings demonstrated that stress could lead to increased IL-6 expression in the DMH and activate JAK/STAT-pathway-related molecules, while significantly reducing GABAAR α1 subunit expression, leading to increased DMH neuronal activity and heart rate. By inhibiting the JAK/STAT pathway, we further demonstrated that the downregulation of GABAAR α1 subunit expression, mediated by the activation of the JAK/STAT pathway, could be an important mechanism that triggers abnormal DMH neuronal activity in stressed rats, thus leading to tachycardia.

## Figures and Tables

**Figure 1 ijms-24-12985-f001:**
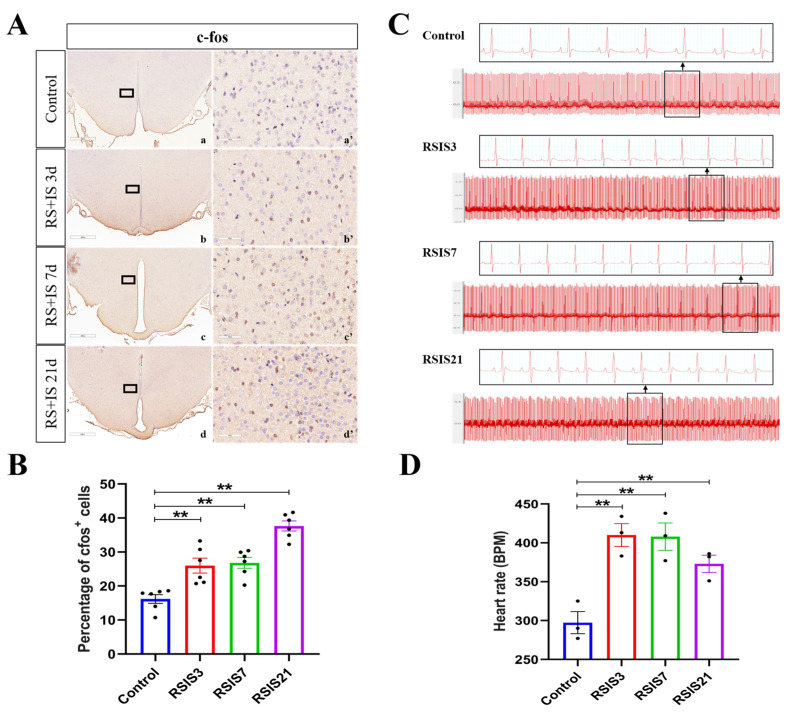
Stress-induced enhancement of DMH neuronal activity increased heart rate. (**A**,**B**) Representative micrographs and quantitative analysis of c-Fos immunohistochemical staining in DMH (n = 6). (**a’**–**d’**) are the magnified areas of (**a**–**d**), respectively. Scale bar = 900 µm in (**a**–**d**); scale bar = 50 µm in (**a’**–**d’**). (**C**) Representative ECG records of rats. (**D**) Changes in heart rate in rats (n = 3). Values are expressed as the mean ± SEM, ** *p* < 0.01 vs. the control group.

**Figure 2 ijms-24-12985-f002:**
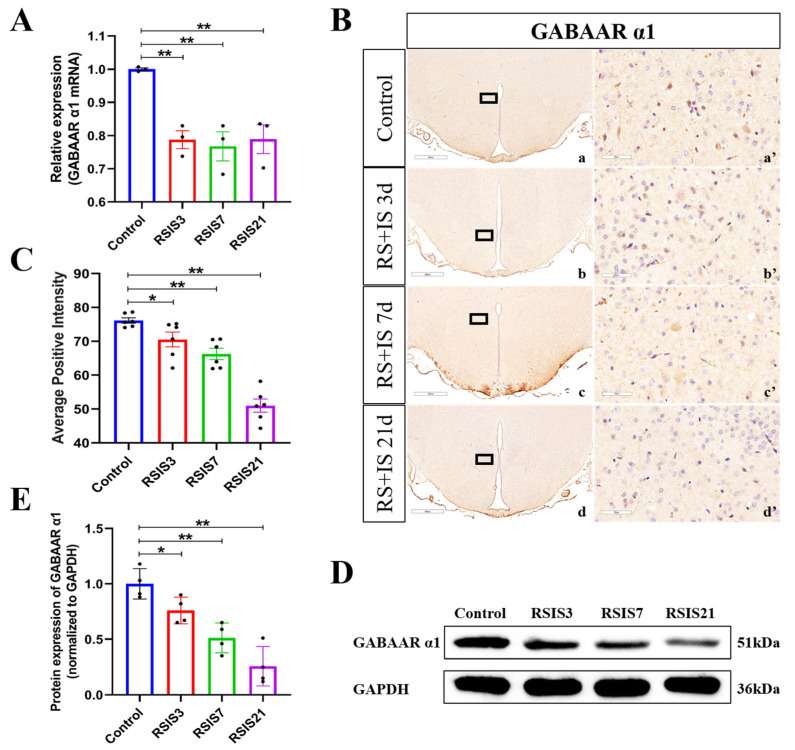
Stress decreased the expression of the GABAAR α1 subunit. (**A**) The mRNA level of GABAA receptor α1 subunit (n = 3). (**B**,**C**) Representative micrographs and quantitative analysis of GABAAR α1 subunit immunohistochemical staining in DMH (n = 6). (**a’**–**d’**) are the magnified areas of (**a**–**d**), respectively. Scale bar = 900 µm in (**a**–**d**); scale bar = 50 µm in (**a’**–**d’**). (**D**,**E**) Representative Western blot images and densitometric quantification of GABAAR α1 subunit in the DMH (n = 4). Values are expressed as the mean ± SEM, * *p* < 0.05, ** *p* < 0.01 vs. the control group.

**Figure 3 ijms-24-12985-f003:**
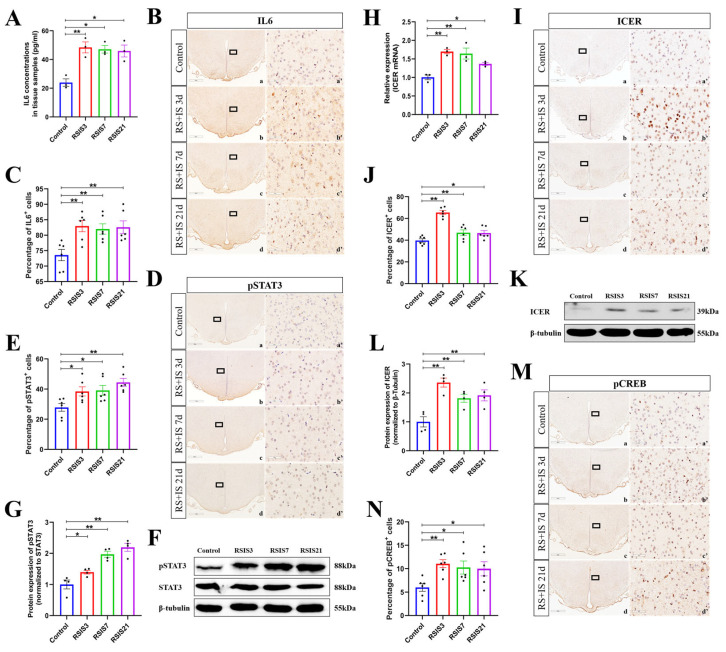
Stress-induced increase in IL-6 activated the JAK/STAT pathway, promoting the expression of the downstream molecule ICER. (**A**) IL6 content in DMH (n = 3). (**B**,**C**) Representative micrographs and quantitative analysis of IL-6 immunohistochemical staining in DMH (n = 6). (**a’**–**d’**) are the magnified areas of (**a**–**d**), respectively. Scale bar = 900 µm in (**a**–**d**); scale bar = 50 µm in (**a’**–**d’**). (**D**,**E**) Representative micrographs and quantitative analysis of pSTAT3 immunohistochemical staining in DMH (n = 6). (**a’**–**d’**) are the magnified areas of (**a**–**d**), respectively. Scale bar = 900 µm in (**a**–**d**); scale bar = 50 µm in (**a’**–**d’**). (**F**,**G**) Representative Western blot images and densitometric quantification of pSTAT3 in the DMH (n = 4). (**H**) The mRNA level of ICER (n = 3). (**I**,**J**) Representative micrographs and quantitative analysis of ICER immunohistochemical staining in DMH (n = 6). (**a’**–**d’**) are the magnified areas of (**a**–**d**), respectively. Scale bar = 900 µm in (**a**–**d**); scale bar = 50 µm in (**a’**–**d’**). (**K**,**L**) Representative Western blot images and densitometric quantification of ICER in the DMH (n = 4). (**M**,**N**) Representative micrographs and quantitative analysis of pCREB immunohistochemical staining in DMH (n = 6). (**a’**–**d’**) are the magnified areas of (**a**–**d**), respectively. Scale bar = 900 µm in (a–d); scale bar = 50 µm in (**a****’**–**d’**). Values are expressed as the mean ± SEM, * *p* < 0.05, ** *p* < 0.01 vs. the control group.

**Figure 4 ijms-24-12985-f004:**
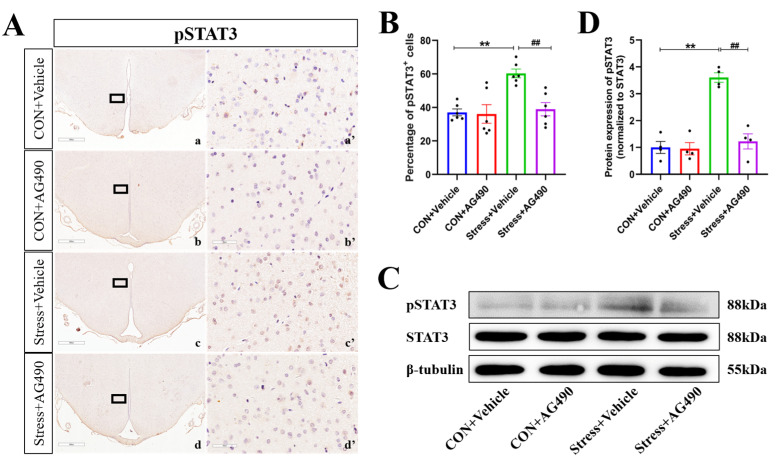
AG490 treatment inhibited the levels of pSTAT3 in stressed rats. (**A**,**B**) Representative micrographs and quantitative analysis of pSTAT3 immunohistochemical staining in DMH after AG490 treatment (n = 6). (**a’**–**d’**) are the magnified areas of (**a**–**d**), respectively. Scale bar = 900 µm in (**a**–**d**); scale bar = 50 µm in (**a’**–**d’**). (**C**,**D**) Representative Western blot images and densitometric quantification of pSTAT3 in the DMH (n = 4). Values are expressed as the mean ± SEM, ** *p* < 0.01, ^##^
*p* < 0.01 vs. the control group.

**Figure 5 ijms-24-12985-f005:**
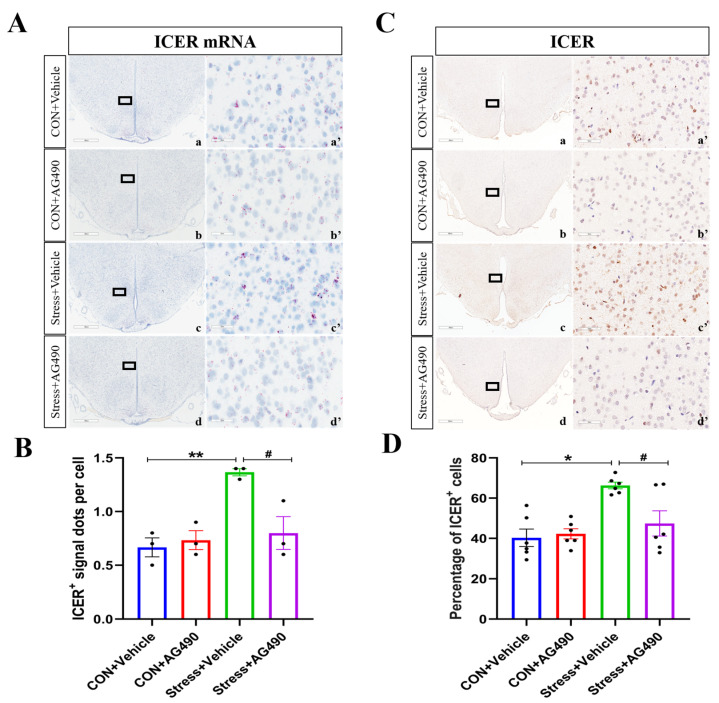
AG490 treatment reduced the expression of ICER in the DMH of stressed rats. (**A**) Representative micrographs of ICER RNAscope in DMH after AG490 treatment. (**a’**–**d’**) are the magnified areas of (**a**–**d**), respectively. Scale bar = 900 µm in (**a**–**d**); scale bar = 50 µm in (**a’**–**d’**). (**B**) The average number of ICER positive signal dots per cell in DMH after AG490 treatment (n = 3). (**C**,**D**) Representative micrographs and quantitative analysis of ICER immunohistochemical staining in DMH after AG490 treatment (n = 6). (**a’**–**d’**) are the magnified areas of (**a**–**d**), respectively. Scale bar = 900 µm in (**a**–**d**); scale bar = 50 µm in (**a’**–**d’**). (**E**,**F**) Representative Western blot images and densitometric quantification of ICER in the DMH (n = 4). Values are expressed as the mean ± SEM, * *p <* 0.05, ** *p <* 0.01, ^#^ *p* < 0.05, ^##^ *p* < 0.01 vs. the control group.

**Figure 6 ijms-24-12985-f006:**
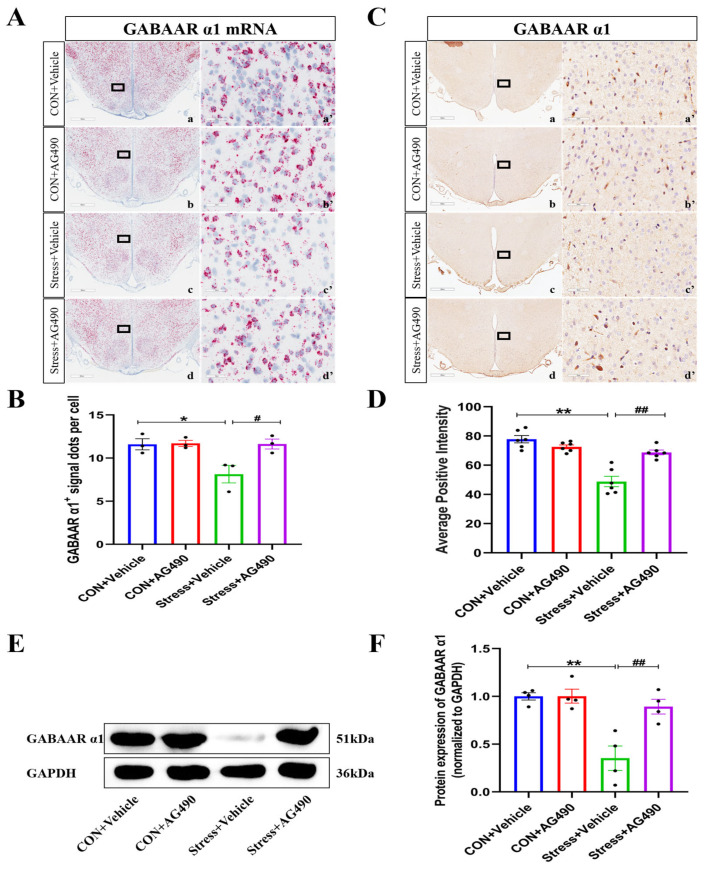
AG490 treatment alleviated the stress-induced decrease in the GABAAR α1 subunit. (**A**) Representative micrographs of GABAAR α1 subunit RNAscope in DMH after AG490 treatment. (**a’**–**d’**) are the magnified areas of (**a**–**d**), respectively. Scale bar = 900 µm in (**a**–**d**); scale bar = 50 µm in (**a’**–**d’**). (**B**) The average number of GABAAR α1 subunit positive signal dots per cell in DMH after AG490 treatment (n = 3). (**C**,**D**) Representative micrographs and quantitative analysis of GABAAR α1 subunit immunohistochemical staining in DMH after AG490 treatment (n = 6). (**a’**–**d’**) are the magnified areas of (**a**–**d**), respectively. Scale bar = 900 µm in (**a**–**d**); scale bar = 50 µm in (**a’**–**d’**). (**E**,**F**) Representative Western blot images and densitometric quantification of GABAAR α1 subunit in the DMH (n = 4). Values are expressed as the mean ± SEM, * *p* < 0.05, ** *p* < 0.01, ^#^ *p* < 0.05, ^##^ *p* < 0.01 vs. the control group.

**Figure 7 ijms-24-12985-f007:**
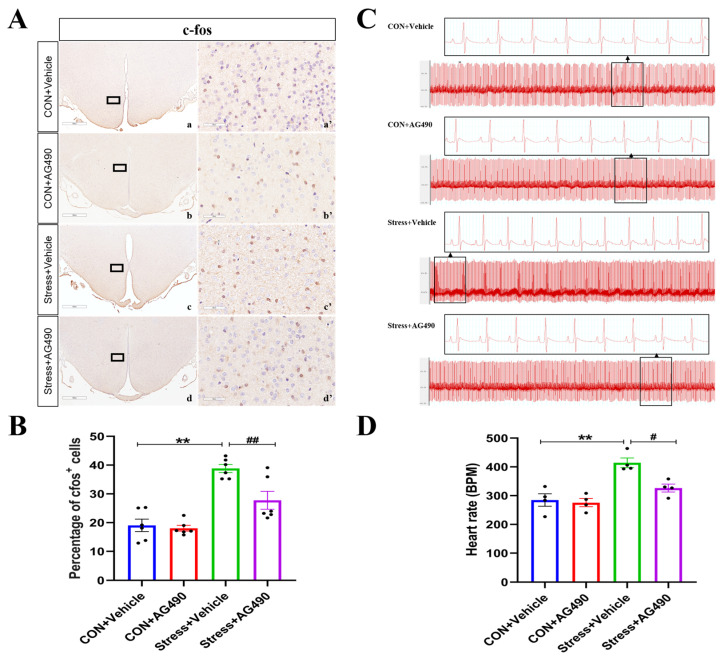
AG490 treatment reduced DMH neuronal activity and improved tachycardia in stressed rats. (**A**,**B**) Representative micrographs and quantitative analysis of c-Fos immunohistochemical staining in DMH after AG490 treatment (n = 6). (**a’**–**d’**) are the magnified areas of (**a**–**d**), respectively. Scale bar = 900 µm in (**a**–**d**); scale bar = 50 µm in (**a’**–**d’**). (**C**) Representative ECG records of rats after AG490 treatment. (**D**) Changes in heart rate in rats (n = 4). Values are expressed as the mean ± SEM, ** *p* < 0.01, ^#^
*p* < 0.05, ^##^
*p* < 0.01 vs. the control group.

## Data Availability

The datasets generated during and/or analyzed during the current study are available from the corresponding author upon reasonable request.

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
