# Peer review of "Interleukin 6 (IL-6) Regulates GABAA Receptors in the Dorsomedial Hypothalamus Nucleus (DMH) through Activation of the JAK/STAT Pathway to Affect Heart Rate Variability in Stressed Rats"

_ijms, 2023, doi:10.3390/ijms241612985_

Round 1
Reviewer 1 Report
The Authors aimed to elucidate the specific molecular mechanisms behind stress leading to abnormal DMH neuronal activity. Therefore, in the present study, they successfully constructed a stressed rat model and used it to investigate the potential molecular mechanisms by which IL-6 regulates GABAA receptors in the DMH through activation of the JAK/STAT pathway, thus affecting also heart rate variability in rats.
To improve the manuscript the authors should consider addressing the following observations.
Fig. 3J:
The percentage of ICER-positive cells shows an increase vs control after RSIS3 returning to a value similar to control after RSIS7 and RSIS21. The Authors should comment on this effect with respect to ICER expression.
AG490 experiments:
- In AG490 experiments (paragraph 2.4), it is not clear which of the three stressing times has been used to test the JAK/STAT pathway. Please specify accordingly.
- In addition, how the Authors decided the amount of AG490 used in related experiments? See line 299 where the authors indicated: (2μl, 5nM, i.c.v.) was dissolved in 3% dimethyl sulfoxide (DMSO).
Author Response
请参阅附件。

Reviewer 2 Report
In the current study, the author explored the mechanism by which hypothalamic DMH neurons affect heart rate variability in stressed rats. The study asked important questions however methodological deficiencies and results presentation damping the overall excitement of the study.
1) Its obvious from the figures panel presented throughout the study that authors compared different rostral to caudal DMH neurons. Additionally, it should be noted that DMH has multiple subdivisions, and it's not clear which DMH subdivision the author is comparing.
2) In IHC, methods are incompletely described, and how the normalization is done is not clear. Even in the figures panels in some instances, overall nuclear staining is different in comparison (control vs stress or drug) than how the normalization is done.
3) There are issues in image brightness and contrast adjustments which makes it believe that results are not appropriately presented.
4) 3% DMSO as the vehicle was injected within 5s directly into i.c.v., It may be cytotoxic to the brain cells.
5) Author should describe the gender used in the study and the timing for each experiment measurement.
Round 2
Reviewer 2 Report
All the issues were addressed.